Sensitivity of deep learning applied to spatial image steganalysis

Tabares-Soto Reinel 1 rtabares@autonoma.edu.co
http://orcid.org/0000-0002-2341-5079 Arteaga-Arteaga Harold Brayan 1
http://orcid.org/0000-0001-6012-8645 Mora-Rubio Alejandro 1
Bravo-Ortíz Mario Alejandro 1
Arias-Garzón Daniel 1
http://orcid.org/0000-0003-1021-2050 Alzate-Grisales Jesús Alejandro 1
http://orcid.org/0000-0001-5991-8770 Orozco-Arias Simon 2 3
http://orcid.org/0000-0002-1089-4605 Isaza Gustavo 3
Ramos-Pollán Raúl 4
1 Department of Electronics and Automation, Universidad Autónoma de Manizales , Manizales, Caldas , Colombia
2 Department of Computer Science, Universidad Autónoma de Manizales , Manizales, Caldas , Colombia
3 Department of Systems and Informatics, Universidad de Caldas , Manizales, Caldas , Colombia
4 Department of Systems Engineering, Universidad de Antioquia , Medellín, Antioquia , Colombia
Alazab Mamoun
Electronic publication date: 2021 Aug 31
Publication date: 2021
Volume: 7
Electronic Location ID: e616
Received 2021 Apr 14; Accepted 2021 Jun 9
Copyright: © 2021 Tabares-Soto et al.
Copyright year: 2021
Copyright holder: Tabares-Soto et al.
License: This is an open access article distributed under the terms of the Creative Commons Attribution License, which permits unrestricted use, distribution, reproduction and adaptation in any medium and for any purpose provided that it is properly attributed. For attribution, the original author(s), title, publication source (PeerJ Computer Science) and either DOI or URL of the article must be cited.
License URL: https://creativecommons.org/licenses/by/4.0/

Keywords: Convolutional neural network, Deep learning, Steganalysis, Steganography, Sensitivity

Funding: Universidad Autónoma de Manizales, Manizales, Colombia 645-2019 TD This work was supported by Universidad Autónoma de Manizales, Manizales, Caldas, Colombia, under project No. 645-2019 TD. The funders had no role in study design, data collection and analysis, decision to publish, or preparation of the manuscript.

==============================
In recent years, the traditional approach to spatial image steganalysis has shifted to deep learning (DL) techniques, which have improved the detection accuracy while combining feature extraction and classification in a single model, usually a convolutional neural network (CNN). The main contribution from researchers in this area is new architectures that further improve detection accuracy. Nevertheless, the preprocessing and partition of the database influence the overall performance of the CNN. This paper presents the results achieved by novel steganalysis networks (Xu-Net, Ye-Net, Yedroudj-Net, SR-Net, Zhu-Net, and GBRAS-Net) using different combinations of image and filter normalization ranges, various database splits, different activation functions for the preprocessing stage, as well as an analysis on the activation maps and how to report accuracy. These results demonstrate how sensible steganalysis systems are to changes in any stage of the process, and how important it is for researchers in this field to register and report their work thoroughly. We also propose a set of recommendations for the design of experiments in steganalysis with DL.

Introduction

In the context of criptography and information hiding, steganography refers to hiding messages in digital multimedia files (Hassaballah, 2020) and steganalysis consists of detecting whether a file has a hidden message or not (Reinel, RaÃol & Gustavo, 2019; Tabares-Soto et al., 2020, Chaumont, 2020). In digital image steganography, a message can be hidden by changing the value of some pixels in the image (spatial domain, see Fig. 1) (Hameed et al., 2019) or by modifying the coefficients of a frequency transform (frequency domain) while remaining invisible to the human eye. Some of the steganographic algorithms in the spatial domain are HUGO (Pevny, Filler & Bas, 2010), WOW (Holub & Fridrich, 2012), S-UNIWARD (Holub, Fridrich & Denemark, 2014), HILL (Li et al., 2014), and MiPOD (Sedighi et al., 2016).

Figure 1 Cover, stego and steganographic content images.

On the other end, there are two main stages in the steganalysis process. Stage one is feature extraction, the most prominent technique being Rich Models (RM) (Fridrich & Kodovsky, 2012), while stage two involves a binary classification model, such as support vector machines (SVM) or perceptrons, to predict if an image is steganographic or not. Nowadays, thanks to the evolution of deep learning (DL) (Theodoridis, 2015) and improvements in computing hardware capabilities (e.g., graphics processing unit (GPU) (Tabares-Soto, 2016) and tensor processing unit (TPU)), unifying both stages under the same model, usually a convolutional neural network (CNN), has become the go-to strategy, improving the detection accuracy of steganographic images. Before the introduction of CNN in steganalysis, the state-of-the-art approaches were spatial rich Models (SRM) (Fridrich & Kodovsky, 2012) and a subtractive pixel adjacency matrix (SPAM) (Pevny, Bas & Fridrich, 2010).

The first CNN architecture applied to steganalysis, with five convolutional layers, a Gaussian activation, and trained using supervised learning, was proposed by Qian et al. (2015). The following year, Xu, Wu & Shi (2016) proposed a CNN with five convolutional layers. The introduction of an absolute value (ABS) layer and 1 × 1 convolutional kernels improved the detection accuracy over Qian et al. (2015) proposal. Despite an improvement in the detection accuracy, these CNN still did not outperform SRM (Fridrich & Kodovsky, 2012) or SPAM (Pevny, Bas & Fridrich, 2010).

It was not until Ye, Ni & Yi (2017) presented a more prominent CNN architecture with eight convolutional layers, that CNN models improved the detection accuracy over the traditional approaches. This network also introduced a novel activation function named Truncation Linear Unit (TLU), and an image-preprocessing layer using filters initialized with SRM-based weights. This approach improved the detection accuracy by approximately 10% compared to the traditional algorithms and previous CNN. Aiming to join the useful elements from earlier proposals, Yedroudj, Comby & Chaumont (2018) proposed a new CNN consisting of: SRM-based filter banks, five convolutional layers, batch normalization (BN), and TLU activation units. Furthermore, the researchers proposed an augmented training database, by adding images from the BOWS 2 database (Mazurczyk & Wendzel, 2017) to the traditional BOSSBase database (Bas, Filler & Pevny, 2011), and by including operations such as cropping, resizing, rotation, and interpolation. In the same year, a CNN able to detect steganographic images in the spatial and frequency domain was proposed by Boroumand, Chen & Fridrich (2019), the main feature of this architecture is the use of residual connections.

Following the most relevant proposals in the steganalysis field, the CNN presented by Zhang et al. (2019) introduced separate convolutions and multi-level average pooling known as Spatial Pyramid Pooling (SPP) (He et al., 2014), which allows the network to process arbitrarily sized images. Tan et al. (2020) sought to decrease the computational cost, storage overheads, and difficulties in training and deployment. The resulting model (i.e., CALPA-Net) improved adaptivity, transferability, and scalability. Furthermore, Wang et al. (2020) proposed a CNN that uses detection mechanisms and joint domains. The authors applied SRM filters and the Discrete Cosine Transform Residual (DCTR) patterns for transformation steganographic impacts.

Currently, GBRAS-Net architecture, presented by Reinel et al. (2021), achieves the highest detection percentages of steganographic images in the spatial domain. In the preprocessing stage, this network keeps the 30 SRM filters and uses a modified TanH activation function. This CNN involves skip-connections, separable and depthwise convolutions using the ELU activation function, for feature extraction. For the classification stage, the CNN uses a softmax directly after global average pooling, removing fully connected layers.

Table 1 shows the performance of the CNN architectures. These results correspond to the most relevant architectures for classifying S-UNIWARD and WOW steganographic images. The payloads used are 0.2 and 0.4 bits per pixel (bpp).

Table 1 Accuracy percentage of models for S-UNIWARD and WOW steganographic algorithms, with payloads of 0.2 and 0.4 bpp.

The bold entries indicate the best results.

Year-Algorithm	S-UNIWARD 0.2 bpp	S-UNIWARD
0.4 bpp	WOW
0.2 bpp	WOW
0.4 bpp	
2020-GBRAS-Net	73.6	87.1	80.3	89.8	
2019-Zhu-Net	71.4	84.5	76.9	88.1	
2018-SR-Net	67.7	81.3	75.5	86.4	
2018-Yedroudj-Net	63.5	77.4	72.3	85.1	
2017-Ye-Net	60.1	68.7	66.9	76.7	
2016-Xu-Net	60.9	72.7	67.5	79.3	
2015-Qian-Net	53.7	69.1	61.4	70.7	
2012-SRM + Ensemble classifier	63.4	75.3	63.5	74.5	

In general, a sensitivity analysis refers to the assessment of how the output of a system, or in this case performance of a model, is influenced by its inputs (Razavi et al., 2021), not only training data, but model hyper-parameters, preprocessing operations, and desing choices as well. Besides assuring the quality of a model (Saltelli et al., 2019), sensitivity analysis can provide an important tool in reporting reproducible results, by explaining the conditions around which those results were achieved (Razavi et al., 2021). In its most simple form, consists of varying each of the inputs around its possible values and evaluating the results achieved.

Given the accelerated growth of DL techniques for steganalysis, measuring how factors such as image and filter normalization, database partition, and activation function can affect the development and performance of algorithms for steganographic images detection is essential. This research was motivated by the lack of detailed documentation of the experimental set-up, the difficulty to reproduce the CNNs, and the variability of reported results. This paper describes the results of a thorough experimentation process in which different CNN architectures were tested under different scenarios to determine how the training conditions affect the results. Similarly, this paper presents an analysis of how researchers can select the products to report, aiming to deliver reproducible and consistent results. These issues are essential to assess the sensitivity of DL algorithms to different training settings and will ultimately contribute to a further understanding of the problems applied to steganalysis and how to approach them.

The paper has the following order: The “Materials and Methods” section describes the database, CNN architectures, experiments, training and hyper-parameters, hardware and resources. The “Results” section presents the quantitative results found for each of the scenarios. The “Discussion” section discusses the results presented in terms of their relationship and effect on steganalysis systems. Lastly, the “Conclusions” section presents the conclusions of the paper.

Materials and Methods

Database

The database used for the experiments was Break Our Steganographic System (BOSSBase 1.01) (Bas, Filler & Pevny, 2011). This database consists of 10,000 cover images of 512 × 512 pixels in a Portable Gray Map (PGM) format (8 bits grayscale). For this research, similar to the process presented by Tabares-Soto et al. (2021), the following operations were performed on the images:All images were resized to 256 × 256 pixels.

Each corresponding steganographic image was created for each cover image using S-UNIWARD (Holub, Fridrich & Denemark, 2014) and WOW (Holub & Fridrich, 2012) with payload 0.4 bpp. The implementation of these steganographic algorithms was based on the open-source tool named Aletheia (Lerch, 2020) and the Digital Data Embedding Laboratory at Binghamton University (Binghamton University, 2015).

The images were divided into training, validation, and test sets. The size of each group varied according to the experiment.

Default partition

After the corresponding steganographic images are generated, the BOSSBase 1.01 database contains 10,000 pairs of images (cover and stego) divided into 4,000 pairs for train, 1,000 pairs for validation, and 5,000 for the test. This partition of the BOSSBase 1.01 database was based on Xu, Wu & Shi (2016), Ye, Ni & Yi (2017), Yedroudj, Comby & Chaumont (2018), Boroumand, Chen & Fridrich (2019), Zhang et al., 2019, Reinel et al. (2021).

CNN Architectures

The CNN architectures used in this research, except for GBRAS-Net, were modified according to the strategy described in Tabares-Soto et al. (2021) to improve the performance of the networks regarding convergence, stability of the training process, and detection accuracy. The modifications involved the following: a preprocessing stage with 30 SRM filters and a modified TanH activation with range [−3, 3], Spatial Dropout before the convolutional layers, Absolute Value followed by Batch Normalization after the convolutional layers, Leaky ReLU activation in convolutional layers, and a classification stage with three fully connected layers (Bravo Ortíz et al., 2021). Figure 2 shows two of the six CNN architectures used for the experiments.

Figure 2 Convolutional neural network architectures.

(A) Xu-Net. and (B) GBRAS-Net.

Complexity of CNNs

There are two dimensions to calculate the computational complexity of a CNN, spatial and temporal. The spatial complexity calculates the disk size that the model will occupy after being trained (parameters and feature maps). The time complexity allows calculating floating-point operations per second (FLOPS) that the network can perform (He & Sun, 2015). Eq. (1) is used to calculate the temporal complexity of a CNN and Eq. (2) is used to calculate the spatial complexity.

(1) Time∼O(∑l=1DMl2.Kl2.Cl−1.Cl)

(2) Space∼O(∑l=1DKl2.Cl−1.Cl+∑l=1DMl2.Cl)

where:

D = number of convolutional network layers (depth)

l = convolutional layer where the convolution process is being performed

Ml = is the size of one side of the feature map in the l − th convolutional layer

Kl = is the size of one side of the kernel applied on the l − th convolutional layer

Cl−1 = number of channels of each convolution kernel at the input of the l − th convolutional layer

Cl = number of convolution kernels at the output of the l − th convolutional layer

It is important to clarify that for spatial complexity, the first summation calculates the total size of the network parameters. The second summation calculates the size of the feature maps. In Table 2, the spatial and temporal complexities of the CNNs worked in this sensitivity analysis can be observed.

Table 2 Spatial and temporal complexity of the CNNs used to perform the steganalysis process.

CNN	Total number of parameters for training	Spatial complexity in megabytes	Temporal complexity in GigaFLOPS	
Xu-Net	87,830	0.45	2.14	
Ye-Net	88,586	0.43	5.77	
Yedroudj-Net	252,459	1.00	12.51	
SR-Net	4,874,074	19.00	134.77	
Zhu-Net	10,233,770	39.00	3.07	
GBRAS-Net	166,598	0.80	5.92	

Experiments

Image normalization

Image normalization is a typical operation in digital image processing that changes the ranges of the pixel values to match the operating region of the activation function. The most used bounds for CNN training are 0 to 255, when the values are integers, and 0 to 1 with floating-point values. The selection of this range affects performance and, depending on the application, one or the other is preferred. The following ranges were tested to demonstrate these effects:[0; 255]: 8 bit integer.

[−12; 8]: minimum and maximum values of the original SRM filters.

[0; 1]: activation function operating region.

[−1; 1]: activation function operating region.

[−0:5; 0:5]: activation function operating region.

SRM filters normalization

As for image normalization, the SRM filter values shown in Fig. 3 impact network performance. To evaluate the effect of different filter values, experiments were performed without normalization and with normalization by a factor of 1/12, which caused filter values to be in the range [−1, 2/3].

Figure 3 The 30 SRM filters values.

Database partition

Dividing the database into three sets is good practice for artificial intelligence applications: the training set to adjust network parameters, the validation set to change network hyper-parameters, and the test set to perform the final evaluation of the CNN performance. There is a default partition (see “Default partition”) which most researchers use in the field. As part of the experimentation process developed in this research, the CNN was tested using three additional database partitions as follow (amounts in image pairs):Train: 2,500, Validation: 2,500, and Test: 5,000.

Train: 4,000, Validation: 3,000, and Test: 3,000.

Train: 8,000, Validation: 1,000, and Test: 1,000.

Activation function of the preprocessing stage

The preprocessing stage, which consisted of a convolutional layer with 30 SRM filters, involves an activation function that affects model performance on specific steganographic algorithms. As part of the experimentation process, four different activation functions were tested: 3 × TanH, 3 × HardSigmoid, 3 × Sigmoid and 3 × Softsign.

Activation maps analysis

The output of a particular layer of a CNN is known as activation maps, indicating how well the architecture performs feature extraction. This paper presents the comparative analysis of the activation maps generated by a cover, a stego, and a “cover-stego” image in a trained model. Furthermore, by comparing, it is possible to see the differences between them.

Accuracy reporting in steganalysis

One of the characteristics of CNN training in steganalysis is the unstable accuracy and loss values between epochs, leading to highly variable results and training curves. Consequently, an abnormally high accuracy value can be achieved at a given time during the training process. Although it is correct to select the best accuracy under comparison, having more data allows a better understanding of the CNN. For example, in this paper, model accuracy was evaluated using the mean and standard deviation of the top five results from training, validation, and testing.

Training and hyper-parameters

The training batch size was set to 64 images for Xu-Net, Ye-Net, Yedroudj-Net, and 32 for SR-Net, Zhu-Net, GBRAS-Net. The number of training epochs needed to reach convergence is 100, except for Xu-Net that uses 150 epochs. The spatial dropout rate was 0.1 in all layers. Batch normalization had a momentum of 0.2, epsilon of 0.001, and renorm momentum of 0.4. The stochastic gradient descent optimizer momentum was 0.95, and the learning rate was initialized to 0.005. Except for GBRAS-Net, all layers used a glorot normal initializer and L2 regularization for weights and bias.

For GBRAS-Net architecture, the training network uses Adam optimizer, which has the following configuration: the learning rate is 0.001, the beta 1 is 0.9, the beta 2 is 0.999, the decay is 0.0, and the epsilon is 1e − 08. Convolutional layers, except the first layer of preprocessing, use a kernel initializer called glorot uniform. CNN uses a categorical cross-entropy loss for the two classes. The metric used is accuracy. Batch Normalization is configured like the other CNNs. In the original network, the maximum absolute value normalizes the 30 high-pass SRM filters for each filter. The same padding is used on all layers. As shown in Fig. 2, the predictions performed in the last part of the architecture directly use a Softmax activation function.

Hardware and resources

As previously described in Tabares-Soto et al. (2021), the architectures and experiment implementations used Python 3.8.1 and TensorFlow (Abadi et al., 2015) 2.2.0 in a workstation running Ubuntu 20.04 LTS as an operating system. The computer runs a GeForce RTX 2080 Ti (11 GB), CUDA Version 11.0, an AMD Ryzen 9 3950X 16-Core Processor, and 128 GB of RAM. The remaining implementations used the Google Colaboratory platform in an environment with a Tesla P100 PCIe (16 GB) or TPUs, CUDA Version 10.1, and 25.51 GB RAM.

GPUs and TPUs enhance deep learning models. Accessing TPUs is done from Google Colaboratory. Once there, the models are adjusted to work with the TPUs. For example, in the GBRAS-Net model, on an 11 GB Nvidia RTX 2080Ti GPU (local computer), one epoch takes 229 s, whereas with the TPU configured in Google Colaboratory, the epoch needs only 52 s. That is, it performs it more than three times faster. For the Ye-Net model, on a 16 GB Tesla P100 GPU (Google Colaboratory), one epoch took 44 s approximately; whereas with the TPU, it only takes 12 s. This verifies that the use of TPU helps the experiments run more efficiently. It is important to note that in Google Colaboratory, we can open several notebooks and use different accounts. Which helps reduce experiment times to an unprecedented level. To achieve a correct training of the CNN, batch sizes must be chosen for each model according to the hardware accelerator employed.

Results

Image normalization

Image normalization is a typical operation in digital image processing that affects the performance of CNN. Different types of normalization processes were performed on the images (cover and stego) of BOSSBase 1.01 with WOW 0.4 bpp. Training and validation were performed with Xu-Net, Ye-Net, Yedroudj-Net, SR-Net, Zhu-Net, and GBRAS-Net CNNs (see Fig. 2 for Xu-Net and GBRAS-Net), with default data partition (see “Default partition”) and no SRM filters normalization.

Table 3 shows the best test accuracy results with different image normalizations in the convolutional neural networks with WOW 0.4 bpp. Figure 4, under the title “Image Normalization” shows the accuracy curves of SR-Net, Zhu-Net, and GBRAS-Net CNNs with WOW 0.4 bpp for different image normalizations.

Figure 4 Accuracy curves for SR-Net, Zhu-Net and GBRAS-Net CNN with WOW 0.4 bpp and image normalization.

(A) 0 to 255. (B) −12 to 8. (C) 0 to 1. (D) −1 to 1. (E) −0.5 to 0.5.

Table 3 Image normalization and best test accuracy for CNNs with WOW 0.4 bpp using BOSSBase 1.01.

The bold entries indicate the best results.

Image
normalization	Test accuracy
Xu-Net [%]	Test accuracy
Ye-Net [%]	Test accuracy
Yedroudj-Net [%]	Test accuracy
SR-Net [%]	Test accuracy
Zhu-Net [%]	Test accuracy
GBRAS-Net [%]	
[0, 255]	82.6	84.8	85.5	84.8	84.2	88.4	
[−12, 8]	78.7	81.6	81.5	83.6	84.9	86.5	
[0, 1]	65.9	72.7	51.0	50.5	78.0	84.4	
[−1,1]	51.4	76.3	52.1	75.9	79.7	84.4	
[−0.5, 0.5]	64.2	76.2	50.6	50.2	77.7	85.1	

SRM filters normalization

The SRM filters have an impact on the performance of CNNs for steganalysis. Therefore, filter normalization was performed by multiplying by 1/12. In Table 4, each image normalization, distribution of classes within each batch of images, and data partition were equal to “Image normalization”; additionally, SRM filter normalization was done by multiplying by 1/12.

Table 4 SRM filters and image normalization and best test accuracy for CNNs with WOW 0.4 bpp using BOSSBase 1.01.

The bold entries indicate the best results.

Image
normalization	Test accuracy
Xu-Net [%]	Test accuracy
Ye-Net [%]	Test accuracy
Yedroudj-Net [%]	Test accuracy
SR-Net [%]	Test accuracy
Zhu-Net [%]	Test accuracy
GBRAS-Net [%]	
[0, 255]	79.7	82.6	81.6	82.8	84.9	87.1	
[−12, 8]	50.8	75.9	52.2	50.4	78.9	83.4	
[0, 1]	50.4	69.6	50.2	81.4	50.0	83.6	
[−1, 1]	50.1	66.8	50.2	81.2	50.8	84.6	
[−0.5, 0.5]	50.2	63.1	50.0	81.5	50.0	85.5	

Table 4 shows the best test accuracy result with a different image and filter normalization in the CNNs with WOW 0.4 bpp. Figure 5, under the title “SRM Filters Normalization” shows the accuracy curves for SR-Net, Zhu-Net, and GBRAS-Net CNNs with WOW 0.4 bpp and a different image and filter normalization.

Figure 5 Accuracy curves for SR-Net, Zhu-Net and GBRAS-Net CNN with WOW 0.4 bpp and image with SRM filter normalization.

(A) 0 to 255. (B) −12 to 8. (C) 0 to 1. (D) −1 to 1. (E) −0.5 to 0.5.

Database partition

In artificial intelligence, the databases are divided into training, validation, and testing. For steganalysis, a default data partition is used (see “Default partition”). Tables 5 and 6 show the best accuracy results, mean accuracy and Standard Deviation (SD) of the best models with different data partitions, image pixel values in the range [0, 255], no SRM filter normalization.

Table 5 Different data partitions and best test accuracy for CNNs with S-UNIWARD 0.4 bppusing BOSSBase 1.01.

Corresponding to train, validation and test, respectively: (A) 4,000, 1,000, 5,000. (B) 2,500, 2,500, 5,000. (C) 4,000, 3,000, 3,000. (D) 8,000, 1,000, 1,000.

CNN	Distribution	Accuracy on test [%]	CNN	Distribution	Accuracy on test [%]	
		Best	Mean	SD			Best	Mean	SD	
Xu-Net	A	79.7	79.0	0.51	SR-Net	A	77.0	76.5	0.32	
	B	78.4	77.3	1.01		B	73.3	73.1	0.23	
	C	79.6	79.4	0.17		C	77.7	77.5	0.20	
	D	85.0	84.4	0.33		D	87.5	87.4	0.14	
Ye-Net	A	81.1	80.5	0.53	Zhu-Net	A	82.6	82.5	0.09	
	B	77.2	76.8	0.41		B	81.2	80.5	0.35	
	C	81.2	80.9	0.21		C	81.2	80.7	0.34	
	D	86.8	86.0	0.63		D	86.9	86.7	0.13	
Yedroudj-Net	A	81.8	81.1	0.47	GBRAS-Net	A	82.8	82.1	0.59	
	B	78.5	77.5	0.91		B	80.8	79.5	1.19	
	C	80.7	80.1	0.51		C	81.7	81.5	0.15	
	D	86.3	85.5	0.63		D	89.1	88.3	0.45	

Table 6 Different data partitions and best test accuracy for CNNs with WOW0.4 bpp using BOSSBase 1.01.

Corresponding to train, validation and test, respectively: (A) 4,000, 1,000, 5,000. (B) 2,500, 2,500, 5,000. (C) 4,000, 3,000, 3,000. (D) 8,000, 1,000, 1,000.

CNN	Distribution	Accuracy on test [%]	CNN	Distribution	Accuracy on test [%]	
		Best	Mean	SD			Best	Mean	SD	
Xu-Net	A	82.6	82.2	0.31	SR-Net	A	85.1	85.0	0.01	
	B	81.4	81.1	0.22		B	84.2	83.5	0.53	
	C	82.8	82.1	0.43		C	84.6	84.4	0.19	
	D	87.3	86.8	0.21		D	89.4	89.1	0.13	
Ye-Net	A	84.8	84.5	0.25	Zhu-Net	A	86.9	86.2	0.32	
	B	82.7	82.2	0.37		B	83.8	83.6	0.16	
	C	83.9	83.3	0.63		C	85.1	84.7	0.24	
	D	88.1	87.7	0.27		D	88.4	88.2	0.14	
Yedroudj-Net	A	85.5	85.1	0.33	GBRAS-Net	A	88.4	87.9	0.34	
	B	84.1	83.5	0.35		B	87.0	86.5	0.35	
	C	85.1	84.6	0.27		C	86.3	86.0	0.24	
	D	88.7	88.4	0.15		D	89.4	89.2	0.13	

Table 5 and Fig. 6 shows the results of the different data partitions with S-UNIWARD 0.4 bpp. Table 6 and Fig. 7 shows the results of the different data partitions with WOW 0.4 bpp.

Figure 6 Boxplots for the S-UNIWARD experiments.

This figure shows different data partitions experiments for novel CNN architectures. Train, Validation, Test: (A) 4,000, 1,000, 5,000. (B) 2,500, 2,500, 5,000. (C) 4,000, 3,000, 3,000. (D) 8,000, 1,000, 1,000.

Figure 7 Boxplots for the WOW experiments.

This figure shows different data partitions experiments for novel CNN architectures. Train, Validation, Test: (A) 4,000, 1,000, 5,000. (B) 2,500, 2,500, 5,000. (C) 4,000, 3,000, 3,000. (D) 8,000, 1,000, 1,000.

Figures 8–10 show the accuracy curves of SR-Net, Zhu-Net, and GBRAS-Net CNN with S-UNIWARD and WOW 0.4 bpp for different data partitions.

Figure 8 Accuracy curves of SR-Net with S-UNIWARD and WOW 0.4 bpp.

This figure shows different data partitions for each row. Train, Validation, Test: (A) 4,000, 1,000, 5,000. (B) 2,500, 2,500, 5,000. (C) 4,000, 3,000, 3,000. (D) 8,000, 1,000, 1,000.

Figure 9 Accuracy curves of Zhu-Net with S-UNIWARD and WOW 0.4 bpp.

This figure shows different data partitions for each row. Train, Validation, Test: (A) 4,000, 1,000, 5,000. (B) 2,500, 2,500, 5,000. (C) 4,000, 3,000, 3,000. (D) 8,000, 1,000, 1,000.

Figure 10 Accuracy curves of GBRAS-Net with S-UNIWARD and WOW 0.4 bpp.

This figure shows different data partitions for each row. Train, Validation, Test: (A) 4,000, 1,000, 5,000. (B) 2,500, 2,500, 5,000. (C) 4,000, 3,000, 3,000. (D) 8,000, 1,000, 1,000.

Activation function of the preprocessing stage

Due to the sensitivity of the model, different modifications, such as changing the activation function, can generate variations in performance. The results achieved by Ye-Net with WOW and S-UNIWARD 0.4 bpp are shown in Table 7. The experiment was performed with a default data partition (see “Default partition”), image pixel values in the range [0, 255], with no SRM filter normalization.

Table 7 Effect of the activation function on two steganographic algorithms (WOW and S-UNIWARD) using Ye-Net architecture, trained on TPU with 200 epochs and batchsize of 64.

Activation function	WOW
(Epoch) accuracy	S-UNIWARD
(Epoch) accuracy	
3 × TanH	(119) 85.0	(196) 83.3	
3 × HardSigmoid	(162) 86.0	(188) 81.8	
3 × Sigmoid	(170) 85.5	(198) 81.8	
3 × Softsign	(154) 85.5	(163) 81.2	

Activation maps for cover, stego, and steganographic noise images

The trained model has an accuracy of 89.8%, with BOSSBase 1.01. The model was implemented with GBRAS-Net and WOW 0.4 bpp, a default data partition (see “Default partition”), image pixel values in the range [0, 255], individual SRM filter normalization.

The activation maps of the first and the three last convolution of the network are shown in Fig. 11. The activation maps correspond to cover, stego, and steganographic content images.

Figure 11 Activation maps of convolutional layers in GBRAS-Net architecture trained with WOW 0.4 bpp.

This figure shows the Input image, the first convolutional layer or pre processing layer with SRM Filters, and the last three convolutional layers.

Figure 12 shows the ROC curves with Confidence Interval (CI) for the WOW steganography algorithm. BOSSBase 1.01 database was used to train the model. These curves correspond to the model presented in Table 1 for GBRAS-Net. The ROC curves show the relationship between the false positive and true positive rates. These curves show the Area Under Curve (AUC) values; higher values indicate that the images were better classified by the computational model, which, in turn, depends on the steganography algorithm and payload.

Figure 12 ROC curves with CI for GBRAS-Net against WOW steganographic algorithm with 0.4 bpp on BOSS Base 1.01.

Accuracy reporting in steganalysis

The results of the experiment are shown with a data distribution consisting of 8,000, 1,000, and 1,000 pairs of images, analyzed in GBRAS-Net and Xu-Net architecture using BOSSBase 1.01, image pixel values in the range [0, 255], with no SRM filter normalization. Table 8 shows the results of accuracy reporting. The model accuracy was evaluated using the mean and standard deviation of the top five results achieved by the CNN during training, validation, and testing.

Table 8 Accuracy report structure.

Show the results with this manner allows understand how themodel have the behavior for a specific experiment.

Xu-Net: Train = 8,000, Valid = 1,000, Test = 1,000		GBRAS-Net: Train = 8,000, Valid = 1,000, Test = 1,000	
S-UNIWARD 0.4 bpp Best 5% Accuracies		S-UNIWARD 0.4 bpp, the best 5% Accuracies	
Train	Epoch	Valid	Test		Train	Epoch	Valid	Test	
83.4	149	77.6	83.3		88.4	99	82.6	87.3	
83.1	144	77.0	85.0		88.1	90	81.2	86.5	
82.9	145	77.4	83.5		87.9	96	81.6	86.8	
82.7	147	76.7	83.4		87.9	86	80.8	85.8	
82.7	148	77.3	83.1		87.8	87	83.0	88.1	
82.9	mean	77.2	83.6		88.0	mean	81.8	86.9	
0.32	standard deviation	0.35	0.77		0.25	standard deviation	0.94	0.85	
Valid	Epoch	Test	Train		Valid	Epoch	Test	Train	
77.6	143	83.9	81.6		83.0	87	88.1	87.8	
77.6	149	83.3	83.4		82.9	94	87.9	87.6	
77.4	145	83.5	82.9		82.9	71	88.3	86.5	
77.4	150	84.1	82.2		82.8	77	88.2	85.8	
77.3	130	84.0	82.3		82.6	99	87.3	88.4	
77.5	mean	83.7	82.5		82.8	mean	87.9	87.2	
0.12	standard deviation	0.36	0.70		0.15	standard deviation	0.42	1.06	
Test	Epoch	Train	Valid		Test	Epoch	Train	Valid	
85.0	144	83.1	77.0		89.1	84	87.2	82.5	
84.4	141	82.7	76.7		88.3	71	86.5	82.9	
84.4	131	82.2	77.3		88.2	77	85.8	82.8	
84.3	133	81.0	76.3		88.1	87	87.8	83.0	
84.2	140	82.2	77.1		87.9	76	82.7	82.2	
84.4	mean	82.2	76.9		88.3	mean	86.0	82.7	
0.33	standard deviation	0.79	0.39		0.45	standard deviation	1.98	0.34	

Discussion

This study presents results obtained from testing different combinations of image and filter normalization ranges, various database partitions, different activation functions for the preprocessing stage, as well as analysis on activation maps of convolutions and how to report accuracy when training six CNN architectures applied to image steganalysis in the spatial domain. The experiments proposed here show highly variable results, indicating the importance of detailed documentation and reports derived from novel work in this field.

Regarding image and SRM filter normalization, as shown in Table 3, the effectiveness of a normalization range depends on the selected CNN, such that SRM normalization (see Table 4) can generate completely different results.

The image normalization experiment demonstrates essential aspects of this analysis. For example, considering the Xu-Net architecture in Table 3, the best result is obtained using images with the original values of the database (i.e., in the range 0 to 255). Given this, one could conclude that there is no need for image normalization in any architecture; however, a different result is observed with the Zhu-Net architecture. Zhu-Net has the best result using the normalization of the pixels from −12 to 8 (inspired by the minimum and maximum values of the original SRM filters). We recommend using the original pixel values as the first option because it is the best option for most of CNN.

When considering the combination of image normalization and filter normalization, the results can be different. For example, for SR-Net architecture from Table 3, the normalization of the pixels between −0.5 to 0.5 generates an accuracy of only 50.2% without filter normalization. Conversely, with normalized SRMs, as shown in Table 4, the SR-Net CNN reaches an accuracy of up to 81.5%. However, as the normalization experiments show, GBRAS-Net is the architecture that best behaves or adapts to changes in data normalization and distributions. We recommend making use of this new architecture.

In the database partition experiment, the architectures’ detection accuracy improved as the training set increased and the test set decreased. Furthermore, if the test dataset reduces considerably, performance on future cases can be affected. In response, recent investigations use the BOWS 2 dataset since it contains more information. Consequently, with a bigger dataset, data partition can have more information on training and test that can enhance performance. A small test set may be an inadequate representation of the distribution of the images that the network must classify in a production setting; thus, a higher detection accuracy with this partition may not lead to a helpful improvement.

Figures 8–10 show that a smaller training set produces highly variable validation and test curves, while a bigger training set generates smoother curves. Furthermore, these curves show how the validation curve can sometimes be higher or lower than the training curve. For this reason, it is better to choose the models from the results obtained in test data. For this reason, a good representation or quantity of test data is also important.

Table 7 shows that using different activation functions implies changes in performance. In Ye-Net for WOW and S-UNIWARD with 3 × TanH, an average accuracy of 84.2% is achieved, and with 3 × HardSigmoid, an average accuracy of 83.9%. Although for WOW, the best result is given by using the 3 × HardSigmoid activation function overall. A model that serves for detection in several steganographic algorithms is better to use 3 × TanH shown by the average value of accuracy.

Figure 11 shows that the activation maps from the stego image have differences with the cover image, which indicates a higher activation of the convolutional layer in the presence of the steganographic noise. Moreover, by comparing the activation maps, it is clear that a good learning process was achieved by extracting relevant features and focusing on borders and texture changes in the images, where the steganographic algorithms are known to embed most of the information. The analysis of the activation maps is an effective tool for researchers to evaluate the learning process and gain an understanding of the features that the CNN recognizes as relevant for the steganalysis task. This shows that GBRAS-Net has an excellent ability to discriminate between images without hidden content and with hidden content.

The design of CNN networks allows capturing steganographic content. The first layer (preprocessing), which contains the filters, is responsible for enhancing this noise while decreasing the content of the input image (see Fig 11. in the Cover and Stego columns for the SRM filters row). The Cover–Stego column in Fig. 11 shows the noise. Adaptive steganography does its job well in adapting to image content; as seen in the image, it does so at hard-to-detect edges and places.

As proposed here (Table 8), the main advantage of accuracy reporting is to be able to determine the consistency of the results based not only on the final value or the best one. To obtain these results, as the architectures are trained, a model is saved from each epoch. With these models, the accuracies are then obtained in the datasets. With this, you can know which are the best models. And with this accuracy reporting mode, when a specific experiment is presented, whoever will reproduce it will see the range of results to expect. As shown here, the sensitivity of deep learning is excellent in this problem, which can lead to reproducing a CNN not obtaining the same result from the reporter.

With all information shown in this work for spatial image steganalysis using deep learning, we propose a set of recommendations for the design of experiments, listed below:Recommendation 1: measure CNN sensitivity to data and SRM filter normalizations.

Recommendation 2: measure CNN sensitivity to data distributions.

Recommendation 3: measure CNN sensitivity to data splits.

Recommendation 4: measure CNN sensitivity to activation functions on preprocessing stage.

Recommendation 5: show activation maps of cover, stego, and steganographic content images.

Recommendation 6: report the top five best epochs with accuracies and their standard deviation.

Finally, the contributions of this paper will be listed at a general level:Sensitivity in the percentages of accuracy in detecting steganographic images when applying different normalizations in the pixels of the images on six architectures of CNNs (see Table 3 and Fig. 4).

Sensitivity in the percentage of accuracy detecting steganographic images when applying different normalizations in the SRM filters in the preprocessing stage on six CNNs architectures (see Table 4 and Fig. 5).

Sensitivity in the percentages of accuracy detecting steganographic images has the partition of the set of images in training, validation, and test (see Tables 5 and 6 and Figs. 8–10).

Sensitivity in the percentages of accuracy detecting steganographic images that have tested different activation functions in the preprocessing stage for the training process (see Table 7).

The importance of analyzing the activation maps of the different convolutional layers to make new designs of CNNs architectures and understand their behavior (see Fig. 11).

The importance of reporting the average and standard deviation in the percentages of accuracy detecting steganographic images to determine the results reported in the experiments (see Table 8).

Some possible limitations of the current work, which was developed under the clairvoyant scenario, come from the nature and characteristics of the database: the use of images with fixed resolutions, the specific cameras used to take the pictures, the bit depth of the images, and that all the experiments were performed in the spatial domain.

Conclusions

As shown by the results presented in this paper, steganalysis detection systems are susceptible to changes in any stage of the process. Factors such as image and filter normalization ranges, database partition, and activation function in the preprocessing stage affect the CNN performance to the point they determine its success. With this in mind, we present the analysis of the activation maps of convolutions for GBRAS-Net as a valuable tool to assess the CNN training process and its ability to extract distinctive features between cover and stego images. Understanding the behavior of steganalysis systems is key to design strategies and computational elements to overcome their limitations and improve their performance. For example, taking Ye-Net as a reference, using the WOW steganographic algorithm with 0.4 bpp, on the BOSSBase 1.01 database and the values of each pixel without any modification (0 and 255), results in an accuracy of 84.8% in the detection of steganographic images, while applying a normalization of the image pixels between 0 and 1 generates a result of 72.7% (see Table 3), taking into account the normal values of the SRM filters (−12 and 8), now if we normalize the values of the previous filters between 0 and 1 with the same characteristics mentioned above, we obtain results of 82.6% and 69.6% respectively (see Table 4). Now with the same CNN and performing different partitions of the data set (training, validation, and test), we observe accuracy results on average between 76.8% and 86.0% for the S-UNIWARD steganographic algorithm with 0.4 bpp (see Table 5). The above and the other results mentioned in this paper highlight the importance of clearly and precisely defining the experiments performed in steganalysis to report the results reliably and facilitate the reproduction of the experiments by the researchers.

Furthermore, we recommend reporting accuracy values as the mean and standard deviation of the top five results, as it helps account for model consistency and reliability. If possible, we encourage researchers to liberate a repository with code and data resources to reproduce the results and report the implementation details thoroughly, taking into account preprocessing and feature extraction techniques, classification process, and hyper-parameters.

Additional Information and Declarations

Competing Interests

Author Contributions

Data Availability

The authors declare that they have no competing interests.

Reinel Tabares-Soto conceived and designed the experiments, performed the experiments, analyzed the data, performed the computation work, prepared figures and/or tables, authored or reviewed drafts of the paper, and approved the final draft.

Harold Brayan Arteaga-Arteaga conceived and designed the experiments, performed the experiments, analyzed the data, performed the computation work, prepared figures and/or tables, authored or reviewed drafts of the paper, and approved the final draft.

Alejandro Mora-Rubio conceived and designed the experiments, performed the experiments, analyzed the data, performed the computation work, prepared figures and/or tables, authored or reviewed drafts of the paper, and approved the final draft.

Mario Alejandro Bravo-Ortíz conceived and designed the experiments, performed the experiments, analyzed the data, performed the computation work, prepared figures and/or tables, authored or reviewed drafts of the paper, and approved the final draft.

Daniel Arias-Garzón conceived and designed the experiments, performed the experiments, analyzed the data, performed the computation work, prepared figures and/or tables, authored or reviewed drafts of the paper, and approved the final draft.

Jesús Alejandro Alzate-Grisales performed the experiments, analyzed the data, performed the computation work, authored or reviewed drafts of the paper, and approved the final draft.

Simon Orozco-Arias analyzed the data, performed the computation work, authored or reviewed drafts of the paper, and approved the final draft.

Gustavo Isaza conceived and designed the experiments, performed the experiments, analyzed the data, performed the computation work, authored or reviewed drafts of the paper, and approved the final draft.

Raúl Ramos-Pollán conceived and designed the experiments, performed the experiments, analyzed the data, performed the computation work, authored or reviewed drafts of the paper, and approved the final draft.

The following information was supplied regarding data availability:

The source code and database information is available at GitHub: https://github.com/BioAITeam/Sensitivity-of-deep-learning-applied-to-Spatial-Image-Steganalysis.

The dataset used to reproduce the results is available at Zenodo: Tabares-Soto, Reinel, Arteaga-Arteaga, Harold Brayan, Mora-Rubio, Alejandro, Bravo-Ortíz, Mario Alejandro, Arias-Garzón, Daniel, Alzate-Grisales, Jesús Alejandro, Orozco-Arias, Simon, Isaza, Gustavo, Ramos-Pollán, Raúl. 2021. Sensitivity of deep learning applied to spatial image steganalysis dataset. DOI 10.5281/zenodo.4884116.

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
