# Peer review of "Sensitivity of deep learning applied to spatial image steganalysis"

_PeerJ Computer Science, doi:10.7717/peerj-cs.616_

## Round 0.1 · original submission · Major Revisions

Dear Dr. Arteaga-Arteaga,

Thank you for your submission to PeerJ Computer Science.
It is my opinion as the Academic Editor for your article - Sensitivity of deep learning applied to spatial image steganalysis - that it requires a number of Major Revisions.

My suggested changes and reviewer comments are shown below and on your article 'Overview' screen.

Reviewer 1 ·

Basic reporting

The language needs polishing and a full revision by a native English speaker is needed.

Experimental design

Methods described with sufficient detail

Validity of the findings

Conclusion should highlight the achievements. Here it is more or less similar to the Abstract.

Additional comments

The paper presents a sensitivity analysis of CNN in steganalysis ((Xu-Net, Ye-Net, Yedroudj-Net, SR-Net, Zhu-Net, and GBRAS-Net)) , and offers recommendations to take into account when performing experiments in image steganalysis in the spatial domain. It is a very important work, since it shows how CNN architectures can be affected by multiple factors, from CNNs in their internal composition to the effect of databases.
I suggest the authors bring the following points into their consideration to revise the manuscript accordingly.

1. Delete the details in caption of Figure 1 as it is already explained in the text.
2. In Table 1. I recommend mentioning what the acronym EC (Ensemble classifier) refers to. Nowhere in the document it is mentioned.
3. In Table 7 it would be interesting to show also some results for 3xTanH using other values to multiply the function, like 5, 8, 13, 21 for example.
4. Show the difference in training times, when using GPU, and when using TPU. This experiment could be presented for GBRAS-Net for example. This would also show the importance of using TPU's.
5. Check, because the text of the link effectively leads to the repository, it cannot be done directly by clicking (possibly due to the generation of the Review PDF because it includes the 353 in the link, but it is still good to verify). It is also appreciated that the repository has been included.
6. In the 'Discussion' section the expression epoch to epoch I do not consider correct, I propose to use each epoch
7. Conclusion should highlight the achievements. Here it is more or less similar to the Abstract.
8. Finally, citing the following will enrich the literature and be very useful for readers:
-Deep learning in steganography and steganalysis, 2020
-Digital Media Steganography:Principles, Algorithms, and Advances, Elsevier, 2020 , ISBN: 9780128194386,
-A Novel Image Steganography Method for Industrial Internet of Things Security, IEEE Transactions on Industrial Informatics, 2021
-An adaptive image steganography method based on histogram of oriented gradient and PVD-LSB techniques,IEEE Access

Reviewer 2 ·

Basic reporting

1. Discuss the main motivations of the current work.
2. List out the man contributions of the current work.
3. Summarize the drawbacks of existing works in the form of a table.4. Some of the recent works such as the following can be discussed in the paper: "Image-Based malware classification using ensemble of CNN architectures (IMCEC), Deep learning and medical image processing for coronavirus (COVID-19) pandemic: A survey, Hand gesture classification using a novel CNN-crow search algorithm".
4. Resolution of the images have to be improved.
5. Compare the current work with recent state-of-the-art.
6. Discuss the limitations of the current work.
7. Present the computational complexity of the current work.

Experimental design

Satisfactory.

Validity of the findings

Satisfactory.

Additional comments

1. Discuss the main motivations of the current work.
2. List out the man contributions of the current work.
3. Summarize the drawbacks of existing works in the form of a table.4. Some of the recent works such as the following can be discussed in the paper: "Image-Based malware classification using ensemble of CNN architectures (IMCEC), Deep learning and medical image processing for coronavirus (COVID-19) pandemic: A survey, Hand gesture classification using a novel CNN-crow search algorithm".
4. Resolution of the images have to be improved.
5. Compare the current work with recent state-of-the-art.
6. Discuss the limitations of the current work.
7. Present the computational complexity of the current work.

Reviewer 3 ·

Basic reporting

no comment

Experimental design

no comment

Validity of the findings

no comment

Additional comments

In this manuscript, Reinel et al applied different CNN architectures to solve an important question faced by spatial image steganalysis, i.e., steganographic images accuracy. The authors first provided an excellent overview of the field and then the working principles behind the different algorithms. An impressive test accuracy was reached, demonstrating the promise of the GBRAS-Net method. Overall, the paper is well written, and I recommend its publication after the authors address the following comments:

1. What is the uncertainty of the CNN’s predictions? Specifically, what is the error bars in accuracies reported in Table 5 and Table 6. And what is the confidence band in accuracy curves (Fig. 4 and Fig. 5).
2. Some numbers in Fig. 4 and Fig. 5 cannot be seen and the figure should be replaced.

3. Although CNN architectures seems to be applied with care, it would be helpful to assign physical meanings to feature and model selections beyond simple black box. More discussions in this aspect will add confidence to the paper.

---

## Round 0.2 · accepted · Accept

Dear Authors,

Congratulations on the acceptance of the article.

Reviewer 2 ·

Basic reporting

This paper presents the results achieved by novel steganalysis networks (Xu-Net, Ye-Net, Yedroudj-Net, SR-Net, Zhu-Net, and GBRAS-Net) using different combinations of image and filter normalization ranges, various database splits, a diverse composition of the training mini-batches, different activation functions for the preprocessing stage, as well as an analysis on the activation maps and how to report accuracy.

Experimental design

Satisfactory

Validity of the findings

Satisfactory

Additional comments

The authors have addressed all the comments. I recommend the paper can be accepted for publication.

Reviewer 3 ·

Basic reporting

'No Comment'

Experimental design

'No Comment'

Validity of the findings

'No Comment'

Additional comments

Please fix text on page 16:

In response, recent investigations use the BOWS 2 dataset since it contains more information and consequently, with a ...